# Cabergoline as a preventive migraine treatment: A randomized clinical pilot trial

**Astrid Johannesson Hjelholt**[1,2,3,4]*, **Flemming Winther Bach**[4,5], **Helge Kasch**[4,5], **Henrik Støvring**[1,4], **Troels Staehelin Jensen**[4], **Jens Otto Lunde Jørgensen**[4,6]

1 Steno Diabetes Center Aarhus, Aarhus University Hospital, Aarhus, Denmark, 2 Department of Clinical Pharmacology, Aarhus University Hospital, Aarhus, Denmark, 3 Department of Clinical Pharmacology, Aalborg University Hospital, Aalborg, Denmark, 4 Department of Clinical Medicine, Aarhus University, Aarhus, Denmark, 5 Department of Neurology, Aarhus University Hospital, Aarhus, Denmark, 6 Department of Endocrinology and Internal medicine, Aarhus University Hospital, Aarhus, Denmark

* ajh@clin.au.dk

## Abstract

### Background

Beneficial effects of dopamine agonist treatment on migraine have been reported but remain to be properly tested. The aim of this study was to examine the effect of cabergoline as preventive treatment for migraine.

### Methods

In a double-blind, placebo-controlled pilot study, 36 adults with ≥ 6 monthly migraine days were enrolled at Aarhus University Hospital. Following a 28-days baseline period, participants were randomized to receive cabergoline 0.5 mg or placebo once weekly for 12 weeks as add-on treatment. An electronic headache diary was completed daily, and headache questionnaires and blood tests were collected at baseline and following the treatment period. Primary outcome was change in monthly migraine days. The trial was registered with ClinicalTrials.gov (NCT05525611).

### Results

Of 101 assessed participants, 36 were enrolled. Baseline monthly migraine days were 13.6 (4.1) in the cabergoline group and 14.0 (5.3) in the placebo group. No significant overall difference in the reduction of monthly migraine days was observed. However, among participants with episodic migraine (n = 20), the mean (SE) reduction in monthly migraine days from baseline to the last 28 days of the treatment period was -5.4 (1.3) with cabergoline compared to -1.8 (0.9) with placebo (p = 0.04) [odds ratio: 0.79 (95% CI 0.65 – 0.95), p = 0.014]. In participants with chronic migraine (n = 13), the reduction in monthly migraine days was not significantly different in the two groups. Patients' global impression of change significantly improved after cabergoline treatment as compared to placebo in the entire group of participants (p = 0.006). The number of participants with episodic migraine achieving ≥ 50% reduction in monthly migraine days tended to increase after cabergoline (p = 0.07). Adverse effects were reported by seven participants on cabergoline and four on placebo, none of which were serious.

**Data availability statement:** Data cannot be shared publicly due to ethical considerations.

The data contain sensitive personal information, but anonymized data can be provided upon reasonable request to the corresponding author or via The Department of Clinical Medicine, Aarhus University for researchers who meet the criteria for access to confidential data and provided that strict data anonymity is secured. For contact information, see: https://clin.au.dk/research/clinical-trial-unit

**Funding:** The author(s) received no specific funding for this work.

**Competing interests:** The authors have declared that no competing interests exist.

## Conclusion

Cabergoline significantly reduced monthly migraine days in episodic migraine without serious adverse effects, supporting further investigation into the use of cabergoline for migraine prevention.

## Introduction

Migraine is a complex neurovascular disorder characterized by recurrent attacks of headache, which are typically unilateral, pulsating and accompanied by nausea, photo- and phonophobia [1]. The estimated global lifetime prevalence is approximately 17.5%, with a higher frequency in women (21.0%) compared to men (11.6%) [2]. According to the Global Burden of Disease 2019 study, migraine is the second leading cause of disability worldwide, quantified as years lived with disability [3]. The mechanisms contributing to the pathophysiology and sexual dimorphism of migraine are not fully understood, but nociceptive signals from the trigeminovascular system in combination with vasoactive compounds such as calcitonin gene-related peptide (CGRP) are considered pivotal [1].

The pharmacological treatment includes acute migraine–specific medication, in particular selective serotonin 5-HT1B and 5-HT1D receptor agonists, as well as preventive treatments aiming to reduce frequency, duration and severity of attacks [1]. Therapies blocking the CGRP pathway have proven effective as both acute and preventive treatment, but the cost is a constraint, and they are not effective in all patients [4].

Ergot alkaloids, which were previously used in migraine treatment, are vasoconstrictors of the carotid artery beds and activate 5-HT as well as dopamine receptors [5]. Growing evidence underscores the importance of dopamine in the pathophysiology of migraine, particularly the dopamine D2 receptor, though its exact role remains ambiguous. Dopamine antagonistic drugs are commonly used for acute migraine treatment and certain premonitory symptoms are considered dopamine-driven [6]. On the other hand, the dopamine D2 receptor agonist bromocriptine, an ergot alkaloid derivative, and other dopamine receptor agonists have demonstrated beneficial effects in migraine management [7,8]. *In vivo* PET imaging studies have shown an imbalance in dopamine D2/D3 receptor activity during migraine attacks, indicating fluctuations in endogenous dopamine release that correlate with the chronicity and frequency of migraine attacks [9]. Additionally, studies in rodents suggest that D2 receptor activation may influence central nociceptive sensitization in chronic migraine [10,11].

A major indication for dopamine agonists, including bromocriptine and cabergoline, is hyperprolactinemia usually caused by a prolactin-secreting pituitary adenoma. Interestingly, hyperprolactinemia is more prevalent in females and often associates with headache, which is frequently alleviated by dopamine agonist treatment [12–15]. In a case series, we observed hyperprolactinemia-associated unilateral headache including migraine to resolve in a rapid and pronounced manner after dopamine-agonist treatment irrespective of adenoma size and prolactin lowering [12]. In line with this, preclinical and clinical studies indicate a link between prolactin and migraine [16]. Prolactin interacts with neural circuits including trigeminal sensory neurons, where it increases neuronal excitability [16]. In addition, moderately elevated prolactin levels are reported in patients with migraine, in whom uncontrolled dopamine agonist-induced prolactin lowering improves headache in a high proportion of patients [17,18]. Cabergoline, which is also a ligand for the 5-HT receptors, including 5-HT1B and 5-HT1D, has fewer side effects compared to other dopamine agonists and its pharmacokinetic properties allow weekly oral administration [19,20].

The aim of this investigator-initiated, randomized, placebo-controlled pilot study was to examine the effect of cabergoline as preventive treatment for migraine.

## Methods

### Study design and participants

This randomized, double-blind, placebo-controlled, parallel trial was conducted at Aarhus University Hospital. Recruitment started on the 5th of September 2022 and ended on the 13th of June 2023. The study comprised two phases: a 28-days baseline period followed by a 12-weeks treatment period. Participants were randomly assigned in a 1:1 ratio to receive either cabergoline 0.5 mg or placebo once weekly as add-on treatment.

The study protocol was approved by the regional Ethics Committee System and the Danish Medicines Agency, conducted in accordance with Good Clinical Practice (GCP) under the supervision by the regional GCP unit, and reported at http://www.clinicaltrials.gov (NCT05525611).

Adults (> 18 years of age) with a history of episodic (< 15 monthly migraine days (MMD)) or chronic migraine (≥ 15 MMD) were enrolled. Migraine was defined in accordance with the International Classification of Headache Disorders, 3rd edition [21]. A minimum of 6 MMD was required. The exclusion criteria encompassed pregnancy, breastfeeding, the use of dopamine antagonistic drugs (including antiemetics such as metoclopramide and domperidone), heart valve disease, severe untreated hypertension, and psychiatric disorders. In addition, fertile women not using safe contraception, patients with planned medication changes, and patients with chronic daily headache including presumed medication overuse headache (MOH) were excluded. Written and oral consent was obtained from all participants.

### Randomization and masking

The pharmacy at Aarhus University Hospital generated the allocation sequence, undertook randomization, blinding, and provision of cabergoline and placebo. Both the cabergoline and placebo were encapsulated to ensure that they were visually identical. Block randomization was used, and participants were randomized in blocks of 12, with an equal 1:1 allocation ratio. Study investigators enrolled participants. Assignment to interventions was based on the allocation sequence. The allocation sequence was stored in sealed, opaque envelopes until the study was unblinded after the last participant's last visit. Both participants and investigators were masked to group assignment. Study data were collected and managed using REDCap (Research Electronic Data Capture) hosted at Aarhus University.

### Procedures

The following information was retrieved daily via an electronic diary during both the baseline and the treatment period: migraine headaches including time of onset and resolution, pain severity (mild, moderate, severe), features (unilateral or bilateral location, pulsating quality, aggravation by physical activity), associated symptoms (nausea, photo- and phonophobia, symptoms of aura), and use of acute migraine-specific and analgesic medications.

At baseline and following the 12-weeks treatment period, the participants completed the Migraine Disability Assessment Score (MIDAS) and the Headache Impact Test-6 (HIT-6) questionnaires. MIDAS assesses migraine-related disability over a 3-month recall period. It contains five questions regarding number of days of missed work/school, reduced productivity at work/school, missed household work, reduced productivity in household work, and missed family and/or social activities. A higher score indicates greater disability. The HIT-6

consists of six items: pain, social functioning, role functioning, vitality, cognitive functioning, and psychological distress. The total HIT-6 score ranges from 36 to 78, with a higher score indicating a greater effect on the daily life.

At the end of the treatment period, the participants reported their impression of change in overall disease status since the start of the study using the single-item Patient Global Impression of Change (PGIC) scale, which uses a 10-point scale ranging from "very much worse" (1) to "no change" (5), and "very much improved" (10). In addition, participants were asked to provide their best guess as to whether they had received active or placebo treatment.

Serum prolactin was measured at baseline and following the 12-weeks treatment phase. At baseline, fertile women underwent a qualitative human chorionic gonadotropin blood test to rule out pregnancy.

## Outcomes

The primary outcome was change in MMD between baseline and the final 28 days of the treatment period. A migraine day was defined as any calendar day on which the participant had onset, continuation, or recurrence of migraine as recorded in the electronic diary. MMD was defined as number of migraine days per 28 days. Secondary outcomes included $\geq$ 50% reduction in MMD, PGIC, use of acute migraine–specific medication (triptans) and change from baseline in MIDAS and HIT-6 score.

Safety was monitored throughout the trial including reporting of adverse events and serious adverse events.

## Statistical analyses

Participants were analysed per protocol as a whole group and stratified into participants with episodic migraine (6-14 MMD at baseline) and participants with chronic migraine (> 15 MMD at baseline). Change in MMD was calculated as the difference between number of migraine days per 28 days at baseline and the final 28 days of diary completion in the treatment period. In case of missing data due to lack of diary completion in the baseline period, a predicted value was imputed by dividing number of migraine days with number of days with diary completion multiplied by 28. The maximum number of missing diary completions in the baseline period was 4 days. The treatment effect was calculated by subtracting MMD change in the placebo group from MMD change in the cabergoline group. Primary and secondary outcomes were analysed using unpaired t-test and presented as mean $\pm$ standard errors (SE), if data were normally distributed. For non-normally distributed data, the Mann Whitney U test was used, and results are reported as medians with interquartile ranges (IQR). Baseline values are presented descriptively as means and standard deviations (SD) for normally distributed variables, or medians with IQR for non-normally distributed variables. To test for normality, Q-Q plots were evaluated in addition to the Shapiro-Wilk normality test. As an additional measure of effect, an unadjusted random effect logistic regression model, accounting for baseline differences in proneness to attacks, was applied using follow-up time as explanatory variable. The estimated odds ratio (OR) denotes the odds of a participant in the cabergoline group having a migraine day, using the placebo group as reference. Binary variables were compared with Fisher's exact test. Correlation analyses were performed using Spearman's rank-order correlation. All statistical analyses were performed using R (version 4.3.0). Results were considered statistically significant at a significance level < 0.05 (two-tailed).

## Results

### Participants

Between September 2022 and February 2023, 101 adults were assessed for eligibility. Fifty-four patients failed to meet inclusion criteria (<6 MMD, comorbidities, breastfeeding, use of a dopamine antagonist, not using anticonception, planned medication changes) and 11declined to participate. Thirty-six participants were included and randomly assigned to cabergoline (n = 18) or placebo (n = 18) all of whom completed the trial. In the final analyses, 3 participants were excluded (1 in the cabergoline group and 2 in the placebo group) due to too few MMD during the baseline period according to the inclusion criteria (Fig 1).

Of the 101 adults assessed for eligibility, 65 were excluded. Thirty-six participants were randomized to receive either cabergoline (n = 18) or placebo (n = 18). All randomized participants completed the trial. Three participants were excluded from the final analyses, due to an insufficient number of migraine days during the baseline period, as specified in the inclusion criteria.

Baseline characteristics are presented in Table 1. Twenty out of 33 participants had episodic migraine, and 13 had chronic migraine. The two treatment groups were comparable except that the age at disease onset was lower in the cabergoline group among participants with chronic migraine. Thirty of the 33 participants were females, and the mean age was 42.9 years with a mean age at migraine onset of 17.2 years. All participants used acute medications, and 30 used migraine–specific medications (triptans). Twenty-two participants were currently using or had previously used migraine-preventive medication; 19 had discontinued use of a preventive medication because of insufficient efficacy or side effects. During the baseline phase, the mean (SD) number of MMD was 13.6 (4.1) in the cabergoline group and 14.0 (5.3) in the placebo group.

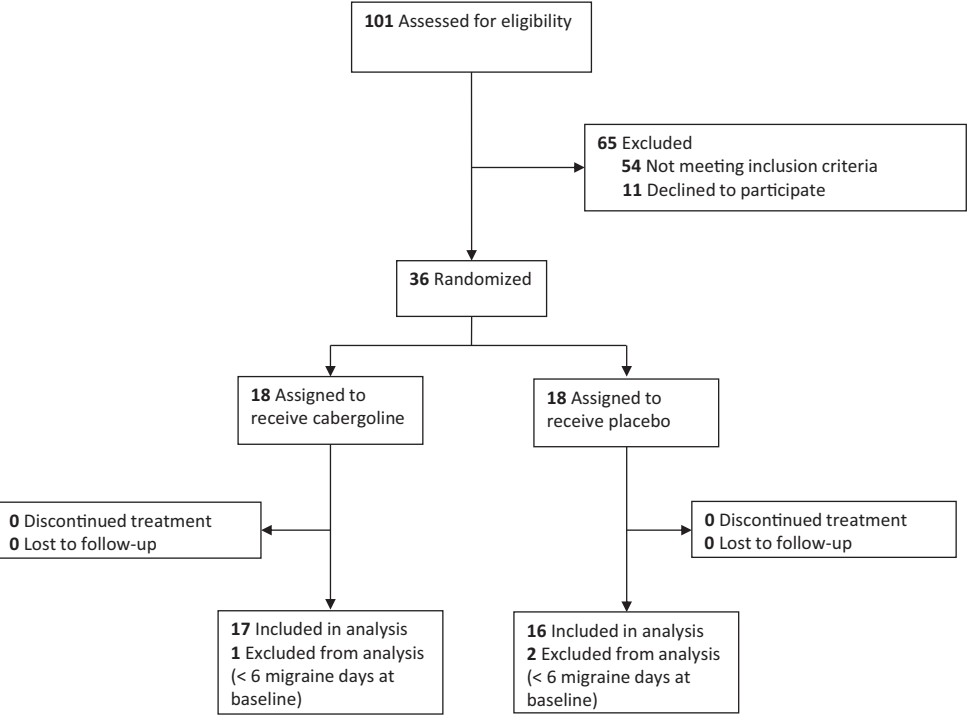

**Fig 1. Flow diagram of the study.**

**Table 1. Baseline Demographic and Clinical Characteristics of the Participants.**

| | Cabergoline | | | Placebo | | |
|---|---|---|---|---|---|---|
| | Total (n = 17) | Episodic migraine (n = 11) | Chronic migraine (n = 6) | Total (n = 16) | Episodic migraine (n = 9) | Chronic migraine (n = 7) |
| Age – years | 41.4 (10.6) | 41.6 (8.9) | 40.8 (14.1) | 44.5 (8.4) | 45.8 (8.3) | 42.9 (8.9) |
| Female sex – no. | 16 | 10 | 6 | 14 | 7 | 7 |
| Age at migraine onset - years | 17.1 (8.7) | 19.9 (9.0) | 12 (5.7)* | 17.2 (6.0) | 14.7 (3.5) | 20.4 (7.2) |
| Acute headache medication use — no. | 17 | 11 | 6 | 16 | 9 | 7 |
| Migraine-specific | 15 | 10 | 5 | 15 | 8 | 7 |
| Non–migraine-specific | 15 | 9 | 6 | 12 | 6 | 6 |
| Migraine-preventive medication use — no. | | | | | | |
| No current or previous use | 6 | 4 | 2 | 7 | 5 | 2 |
| Previous use only | 4 | 4 | 0 | 3 | 1 | 2 |
| Current use | 7 | 3 | 4 | 6 | 3 | 3 |
| History of preventive treatment failure — no. | 10 | 6 | 4 | 9 | 4 | 5 |
| Lack of efficacy | 7 | 4 | 3 | 6 | 3 | 3 |
| Unacceptable side effects | 8 | 4 | 4 | 7 | 3 | 4 |
| Assessment of migraine during baseline phase (per 28 days) | | | | | | |
| Migraine days | 13.6 (4.1) | 11.0 (2.0) | 18.3 (2.0) | 14.0 (5.3) | 10.6 (2.6) | 18.3 (4.6) |
| Days of use of acute migraine medication | 10.5 (4.5) | 8.3 (2.7) | 14.7 (4.4) | 11.5 (3.1) | 9.5 (2.5) | 14.1 (1.1) |
| HIT-6 score | 64 (5.6) | 63.2 (6.6) | 65.3 (3.4) | 63.8 (3.4) | 63.6 (3.8) | 64.1 (3.1) |
| MIDAS grade | 47 (27.5, 85) | 34 (23, 47) | 71.5 (50.5, 143) | 43 (27, 58.5) | 39 (22.8, 65.2) | 47 (31, 52) |

Plus–minus values are means (SD) or medians (IQR). *p < 0.05

## Change in number of monthly migraine days

In the entire group of participants, no difference in MMD change between cabergoline and placebo was found [median (IQR): -5 (-9, -3) days (cabergoline) vs. -3.7 (-5.5, -1.5) days (placebo), p = 0.4]. Subgroup analyses were conducted on participants with episodic and chronic migraine. In participants with episodic migraine, cabergoline significantly reduced MMD as compared to placebo [mean (SE): -5.4 (1.3) (cabergoline) vs. -1.8 (0.9) (placebo); mean (95% CI) difference from placebo was -3.6 (-7.0, -0.2) (p = 0.04)]. In participants with chronic migraine, there was no significant change in number of MMD during cabergoline treatment as compared to placebo [median (IQR): -6.5 (-8.75, -1,25) days vs. -5 (-13, -4) days, (p = 0.6)] (Table 2, Fig 2).

Among participants with episodic migraine, the OR of having a migraine day after cabergoline treatment as compared to placebo was 0.79 (95% CI: 0.65-0.95, p = 0.014), while participants with chronic migraine had an OR of 1.3 (95% CI: 0.87-1.91, p = 0.2).

## Fifty percent reduction in monthly migraine days

A 50% or greater reduction in MMD was achieved for 8 out of 17 participants in the cabergoline group (47%), as compared with 4 out of 16 in the placebo group (25%) (p = 0.28). In participants with episodic migraine, 6 out of 11 participants in the cabergoline group (55%) and 1 out of 9 participants in the placebo group (11%) obtained a reduction of at least 50% (p = 0.07), whereas 2 out of 6 participants with chronic migraine in the cabergoline group (33%) and 3 out of 7 in the placebo group (47%) had a reduction of 50% or greater (Fig 3).

**Table 2. Primary and Secondary Endpoints.**

| | Cabergoline | | | Placebo | | |
|---|---|---|---|---|---|---|
| | Entire group (n = 17) | Episodic migraine (n = 11) | Chronic migraine (n = 6) | Entire group (n = 16) | Episodic migraine (n = 9) | Chronic migraine (n = 7) |
| **Primary endpoint** | | | | | | |
| MMD | | | | | | |
| Change from baseline | -5 (-9, -3) | -5.4 (1.3) | -6.5 (-8.8, -1,3) | -3.7 (-5.5, -1.5) | -1.8 (0.9) | -5 (-13, -4) |
| Difference vs placebo (95% CI) | -1.3 (-5.5, 3) | -3.6 (-7.0, -0.2)* | -1.5 (-8.3, 3) | – | – | – |
| Odds ratio (95% CI) | 1 (0.80, 1.26) | 0.79 (0.65, 0.95)* | 1.3 (0.87, 1.91) | – | – | – |
| **Secondary endpoints** | | | | | | |
| ≥50% Reduction from baseline in MMD | | | | | | |
| No. of patients | 8/17 | 6/11 | 2/6 | 4/16 | 1/9 | 3/7 |
| PGIC | 8 (6, 9) * | 8 (6, 8.5) * | 7.5 (6.3, 8.8) | 5 (5, 6.3) | 5 (5, 5) | 5 (5, 7) |
| Change in acute migraine-specific medication from baseline | -3 (-7, 0) | -3 (-7, 0) | -2 (-4.8, 0.8) | -2.5 (-4.5, -0.8) | -1 (-1, 0) | -4 (-8, -3.5) |
| Change in HIT-6 score from baseline | -3.8 (1.7) | -2.8 (2.5) | -5.2 (2.24) | -2.6 (1.1) | -2.0 (1.2) | -3.4 (1.9) |
| Change in MIDAS score from baseline | -9 (-27.5, -2.8) | -2.5 (-9, 15.2) | -35.5 (-62, -9.8) | -10 (-32.5, -2.5) | -8.5 (-27.5, -0.8) | -18 (-32, -6.5) |
| Change in serum prolactin | -112 (-183, -95)* | -130 (105)* | -106 (80.2)* | -0.5 (-44.8, 37.8) | 0 (57) | -1 (84.5) |

Plus–minus values are means (SE) or median (IQR). Patients' Global Impression of Change (PGIC). * p < 0.05, cabergoline vs. placebo.

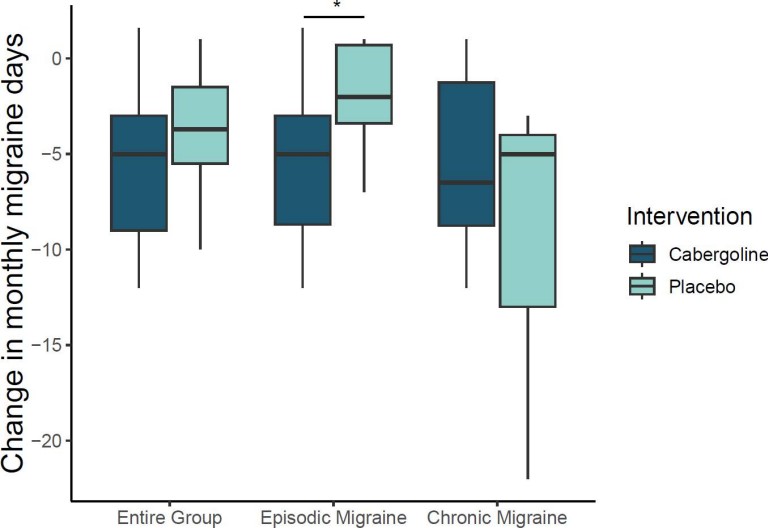

**Fig 2. Change in monthly migraine days (MMD) by subgroup.** Change in number of monthly migraine days (MMD) from baseline to the final 28 days of the treatment period for the entire group, participants with episodic migraine, and participants with chronic migraine. Medians, percentiles, minimums, and maximums are presented. * p < 0.05.

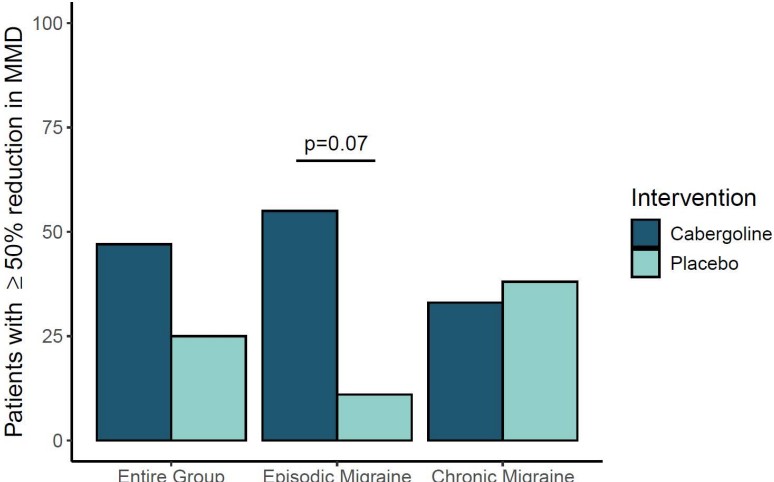

**Fig 3. The proportion of participants achieving ≥ 50% reduction in monthly migraine days (MMD).** Percentage of participants achieving a ≥ 50% reduction in monthly migraine days (MMD) in the entire group of participants, in participants with episodic migraine and in participants with chronic migraine.

## Patient Global Impression of Change (PGIC)

PGIC significantly improved after cabergoline compared to placebo in the entire group [median (IQR): 8 (6, 9) vs. 5 (5, 6.3), p = 0.006]; sub analysis revealed that the PGIC improvement was predominant in participants with episodic migraine [median (IQR): 8 (6, 8.3) vs. 5 (5, 5), p = 0.01] as compared to in chronic migraine [median (IQR): 7.5 (6.3, 8.8) vs. 5.5 (5, 7), p = 0.2] (Fig 4). Four out of 17 participants in the cabergoline group and 11 out of 16 in the placebo group reported *no change* or *change for the worse* after treatment (p = 0.01). The PGIC score correlated inversely with the change in MMD in all participants (r = -0.67, p < 0.0001), and in participants with episodic migraine (r = -0.71, p = 0.0005) and chronic migraine (r = -0.68, p = 0.01).

## Change from baseline in MIDAS and HIT-6 score and use of acute migraine–specific medication (triptans)

No differences in triptan use or change in MIDAS or HIT-6 score were found between groups (Table 2).

## Participants' best guess about treatment allocation

Twelve out of 16 participants in the cabergoline group correctly guessed active treatment as compared to 3 out of 15 in the placebo group (p = 0.004). In participants with episodic migraine, 8 out of 11 in the cabergoline group correctly guessed active treatment as compared to 1 out of 9 in the placebo group (p = 0.01). In participants with chronic migraine, 4 out of 6 in the cabergoline group and 2 out of 7 in the placebo group guessed they had received the active treatment (p = 0.2).

## Serum prolactin measurements

All participants had serum prolactin levels within the normal range and prolactin levels declined significantly after cabergoline (Table 2). Participants with chronic migraine had higher prolactin levels (mU/l) at baseline as compared to participants with episodic migraine [median

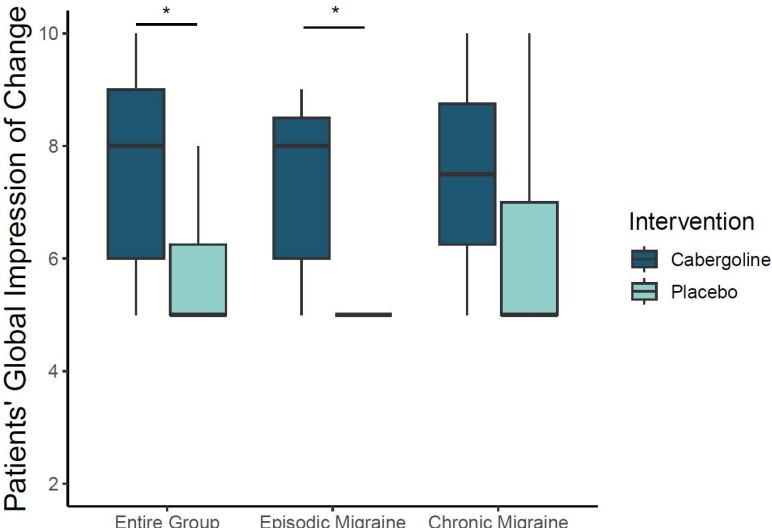

**Fig 4. Patients' Global Impression of Change (PGIC).** Patients' Global Impression of Change (PGIC) in the entire group of participants, in participants with episodic migraine and in participants with chronic migraine. Medians, percentiles, minimums, and maximums are presented. *p < 0.05.

(IQR): 204 (169, 260) vs 276 (220, 363), p = 0.02], whereas there was no significant difference at follow up [median (IQR): 158 (82.5, 201) vs 181 (94, 293), p = 0.2]. Prolactin levels tended to correlate with MMD at baseline (r = 0.3, p = 0.09). No correlation between change in prolactin and change in MMD was found in participants with either episodic or chronic migraine.

## Adverse events

Seven participants in the cabergoline group and 4 in the placebo group experienced adverse effects (p = 0.05) (Table 3). The most common (≥ 2 participants) adverse events in the cabergoline group included headache the day following medication, fatigue, dizziness, and obstipation, while headache following medication was the most frequent adverse event in the placebo group. In general, participants reported that the adverse events subsided during the treatment period. No serious adverse events were reported.

## Discussion

In this pilot study, we tested the effect of preventive cabergoline treatment in a weekly dose of 0.5 mg in migraine. While the primary outcome was not achieved, cabergoline significantly reduced MMD in episodic migraine and led to a significant global patient perceived improvement in the entire group of participants. These beneficial effects of preventive dopamine agonist treatment in migraine substantiate observations from uncontrolled studies [12,13,15,18].

Among participants with chronic migraine, the reduction in MMD was not significant, which may relate to inadequate study power or a true lack of treatment efficacy in these patients. Our study population comprised only 13 participants with chronic migraine, which is noteworthy since this group commonly exhibits a lower treatment response [22] also to CGRP-targeting therapies [23], but treatment responses in chronic migraine may vary, underscoring the need for further research in this subgroup. Migraine trials are characterized by a substantial placebo effect [24] and the larger placebo response in participants with chronic migraine in our study may be the combination of more severe symptoms and higher expectation of pain relief [25].

**Table 3. Adverse Events Reported during the Treatment Phase.**

|  | Cabergoline (n = 18) | Placebo (n = 18) |
| --- | --- | --- |
| Adverse event – no. | 7 | 4 |
| Headache the day following medication | 4 | 2 |
| Fatigue | 4 | – |
| Dizziness | 2 | – |
| Obstipation | 2 | – |
| Sweating | 1 | – |
| Stabbing chest pain | 1 | – |
| Weight gain | 1 | – |
| Headache | – | 1 |
| Loss of appetite | – | 1 |

More than one adverse event could be reported by a participant.

When comparing the outcome of this study to contemporary migraine trials, the efficacy of cabergoline in episodic migraine in terms of a MMD reduction of 3.6 days appears clinically meaningful [26]. According to a recent review, the average reduction in MMD for anti-CGRP treatments versus placebo in episodic and chronic migraine were 1.9 and 2.2 days respectively, while reductions by other preventive treatments, including propranolol, metoprolol, onabotulinumtoxinA, topiramate, valproate, candesartan and amitriptyline, ranged from 0.9 to 1.7 days in episodic migraine and 1.8 to 2 days in chronic migraine [26]. It is important to interpret such comparisons with caution, as the populations in these studies likely differed in baseline characteristics, nevertheless, the cabergoline effect observed in episodic migraine is promising and warrants further investigation in a larger trial.

Next to lack of efficacy, adverse events are the most common reason for discontinuation of preventive migraine treatment [26]. Cabergoline was well-tolerated, with side effects limited to mild and transient symptoms, including headache, dizziness, fatigue, and constipation. Notably, PGIC improved in the treatment group, indicating that patients perceived the benefits of treatment to outweigh any adverse effects experienced. Although the major mechanism of action for cabergoline is via the D2 receptor, cabergoline is also a 5-HT receptor ligand, which could introduce a risk of MOH similar to what is reported after long-term frequent use of triptanes [27]. This merits consideration in any future trial.

While long-term, high-dose cabergoline treatment (> 3 mg/day) in patients with Parkinson's disease has been associated with an increased risk of fibrotic adverse events, including valvular heart disorder, the British Society of Echocardiography, the British Heart Valve Society, and the Society for Endocrinology, concluded that evidence suggesting low-dose dopamine receptor agonist treatment to be associated with abnormal valve morphology or dysfunction is extremely limited, and there is no evidence of clinically significant valve pathology, aside from isolated case reports with unresolved questions [28]. This is corroborated by recent studies including hard cardiac endpoints, which conclude that data do not support an association between low-dose cabergoline treatment and clinically significant valvulopathy [29] or other fibrotic adverse events [30]. In the present study, the cabergoline dose was similar to that used for the treatment of hyperprolactinemia with no serious adverse events reported.

Dopamine has been implicated in the pathophysiology of migraine, but its exact role remains elusive [6,31]. On one hand, dopaminergic symptoms, such as yawning, nausea, and gastrokinetic disturbances occur in the premonitory phase of migraine, and $D_2$-like receptor antagonists may provide relief of these symptoms [6,32]. On the other hand, preclinical studies in rats indicate that dopamine attenuates nociceptive signalling in the trigeminocervical

complex, which expresses dopamine receptors, and plays a central role in migraine patho-physiology [33]. In line with this, clinical data suggest that dopamine agonists may exert beneficial effect in the headache phase of migraine [6–8]. Of particular interest, continuous bromocriptine treatment for one year reduced migraine frequency by 72% in 18 of 24 women with menstrual migraine [7]. The study was, however, uncontrolled and provided limited insight into underlying mechanisms [7].

As mentioned, dopamine agonists including bromocriptine and cabergoline are used to treat hyperprolactinemia, and it has been suggested that prolactin *per se* plays a role in migraine pathophysiology [16,34,35]. Excursions in serum prolactin are hypothesized to trigger menstrual migraine and preclinical data demonstrate prolactin induced migraine-like behaviours and increased excitability in trigeminal neurons in female rodents [36,37]. Prolactin receptors, which are distributed in the trigeminal pain pathway [38], exist as long and short receptor isoforms. Whereas the short isoform induces neuronal excitability and hyperalgesia, the long isoform has been associated with pain protection [36]. Interestingly, cabergoline treatment prevented pain via mechanisms involving prolactin lowering and up-regulation of the long prolactin receptor isoform in female mice [36]. Moreover, prolactin receptor antagonism attenuated dural CGRP-induced migraine-like behaviours. On the other hand, it is also argued that elevated prolactin levels observed in patients with migraine could be an epiphenomenon of reduced dopamine activity, and headache relief after dopamine ago-nist treatment in patients with hyperprolactinemia does not correlate with prolactin lowering [12]. In the present study, all participants had serum prolactin levels within the normal range at baseline, although higher levels were found in chronic migraine. Cabergoline significantly decreased prolactin levels in all participants, indicating compliance, but this was not cor-related to a reduction in MMD.

Evidence from clinical studies suggests that impaired brain glucose metabolism and mito-chondrial dysfunction, leading to a mismatch between the brain's energy supply and work-load, contribute to migraine pathophysiology. Moreover, elevated inflammatory markers are frequently observed in individuals with migraine, indicating a state of chronic inflammation that may exacerbate migraine symptoms [39,40]. This is of particular interest, as cabergoline treatment in patients with prolactinoma has been shown to significantly enhance insulin sen-sitivity and reduce inflammatory markers, independent of its effects on prolactin, LDL cho-lesterol, and BMI [41,42]. Given the design of the current study, it is not possible to determine whether improvements in insulin sensitivity and reductions in inflammation contribute to the beneficial effects of cabergoline in migraine patients. These potential mechanisms, however, should be explored in future studies.

The major strength of our study lies in its the randomized, placebo-controlled design, which minimizes bias and enhances validity. However, the small sample size remains a limita-tion, increasing the risk of random variability and the potential of overestimation or under-estimation of effects. Subgroup analyses in smaller populations are particularly susceptible to spurious findings, underscoring the importance of cautious interpretation of these results. Additionally, the subgroup analysis in patients with chronic and episodic migraine was not predefined, however it was justified by its clinical relevance. Finally, the predominance of female participants limits the generalizability of the results, and our findings should therefore be interpreted with caution when applied to male patients. A larger study is needed to sub-stantiate our findings. In this context, a longer follow-up period may be considered to evaluate long-term efficacy, safety, and durability of cabergoline's effects.

We found no evidence of untoward unblinding, and the participants' guess did not seem driven by adverse effects. The placebo effect in our study is low compared to other preventive migraine trials [24], which speaks against increased expectation bias.

There is still an unmet need for more effective and well-tolerated preventive treatments of patients with episodic and chronic migraine. At the same time, the cost of migraine pharmacotherapies has increased dramatically, mainly due to the increased use of CGRP-targeted compounds [43]. In this context, repurposing of cabergoline would be an affordable option.

In conclusion, this study indicates that cabergoline may be effective in the preventive treatment of migraine without major adverse effects. These results merit to be evaluated in a larger randomized controlled trial.

## Supporting information

**S1 File. CONSORT checklist.**
(DOC)

**S2 File. Study protocol.**
(DOCX)

## Acknowledgement

We would like to thank Pia Buchtrup Hornbek and Lenette Egelund Pedersen from the Medical/Steno Aarhus Research Laboratory, Aarhus University, for their lab assistance, and the study participants for their valuable contributions.

## Author contributions

**Conceptualization:** Astrid Johannesson Hjelholt, Flemming Winther Bach, Helge Kasch, Troels Staehelin Jensen, Jens Otto Lunde Jørgensen.

**Data curation:** Astrid Johannesson Hjelholt.

**Formal analysis:** Astrid Johannesson Hjelholt, Henrik Støvring.

**Funding acquisition:** Jens Otto Lunde Jørgensen.

**Investigation:** Astrid Johannesson Hjelholt.

**Methodology:** Astrid Johannesson Hjelholt, Flemming Winther Bach, Jens Otto Lunde Jørgensen.

**Project administration:** Astrid Johannesson Hjelholt.

**Resources:** Flemming Winther Bach, Jens Otto Lunde Jørgensen.

**Supervision:** Flemming Winther Bach, Helge Kasch, Troels Staehelin Jensen, Jens Otto Lunde Jørgensen.

**Visualization:** Astrid Johannesson Hjelholt.

**Writing – original draft:** Astrid Johannesson Hjelholt, Jens Otto Lunde Jørgensen.

**Writing – review & editing:** Astrid Johannesson Hjelholt, Flemming Winther Bach, Helge Kasch, Henrik Støvring, Troels Staehelin Jensen, Jens Otto Lunde Jørgensen.

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
