## [Decision Letter · Decision Letter 0]

23 Dec 2024

PONE-D-24-49788Cabergoline as a preventive migraine treatment

A randomized clinical pilot trialPLOS ONE

Dear Dr. Hjelholt,

Thank you for submitting your manuscript to PLOS ONE. After careful consideration, we feel that it has merit but does not fully meet PLOS ONE’s publication criteria as it currently stands. Therefore, we invite you to submit a revised version of the manuscript that addresses the points raised during the review process. **I would like to thank the authors for their efforts. We have invited more than usual number of reviewers (3-4) because 2 of the first 3 reviewer reports recommended a rejection and we wanted to give the study a fair shot. Please give attention to all comments and handle as possible, with proper addressing, the paper can move forward.**

We look forward to receiving your revised manuscript.

Kind regards,

Sherief Ghozy, M.D.

Academic Editor

PLOS ONE

**Journal Requirements:**

2. The full date for the end of recruitment is missing from the methodology, in the revised manuscript please include the full date of the end of recruitment including the year.

Reviewers' comments:

Reviewer's Responses to Questions

**Comments to the Author**

1. Is the manuscript technically sound, and do the data support the conclusions?

Reviewer #1: Partly

Reviewer #2: No

Reviewer #3: Yes

Reviewer #4: Yes

Reviewer #5: Yes

Reviewer #6: Yes

Reviewer #7: No

Reviewer #8: Yes

Reviewer #9: Yes

2. Has the statistical analysis been performed appropriately and rigorously? 

Reviewer #1: Yes

Reviewer #2: No

Reviewer #3: I Don't Know

Reviewer #4: Yes

Reviewer #5: I Don't Know

Reviewer #6: Yes

Reviewer #7: No

Reviewer #8: Yes

Reviewer #9: Yes

3. Have the authors made all data underlying the findings in their manuscript fully available?

Reviewer #1: No

Reviewer #2: Yes

Reviewer #3: Yes

Reviewer #4: No

Reviewer #5: Yes

Reviewer #6: Yes

Reviewer #7: Yes

Reviewer #8: Yes

Reviewer #9: Yes

4. Is the manuscript presented in an intelligible fashion and written in standard English?

Reviewer #1: Yes

Reviewer #2: Yes

Reviewer #3: Yes

Reviewer #4: Yes

Reviewer #5: Yes

Reviewer #6: Yes

Reviewer #7: Yes

Reviewer #8: Yes

Reviewer #9: Yes

5. Review Comments to the Author

**Reviewer #1: ** The sample size is extremely small and so the study is not qualified as a randomized trial. The results are hardly generalizable. Also results are dominated for female subjects.

Yet from Table 3 we can see a large number of adverse events in the Cabergoline group compared to control group, even for this small study. The drug may not be safely applied.

The results are full of uncertainty. For example, the box plots in Figure 2 are very wide. The chronic migraine subjects even show smaller changes for placebo group than cabergoline group.

**Reviewer #2: ** This is a double-blind, placebo-controlled study that investigated the effect of cabergoline 0.5mg for migraine prevention. There are several key issues with this trial that need to be explained.

1. According to the clinicaltrial.gov online registry, this is supposed to be a study of chronic migraine, which is defined as having more of than 15 headache days per month, of which at least 8 are migrainous headaches. Therefore, the inclusion criteria clearly contradict the topic of this registration, and the current study includes both episodic and chronic migraine.

2. There is insufficient evidence to justify the use of cabergoline as a migraine preventive drug, given the potential side effects of gambling/sex/other urges, and others. The authors did not provide sufficient argument for its use in migraine other than case series of comorbid migraine and hyperprolactinemia. The potential binding affinity of cabergoline to 5-HT1B/1D receptors is not a valid argument, as studies with triptans (5-HT1B/1D agonists) do not provide evidence of prevention but may increase the risk of medication overuse headache.

3. The study cohort is small (18 cabergoline vs. 18 placebo). There is no information on the power calculation and whether the study is sufficiently powered to detect a difference between the two cohorts.

4. The primary outcome registered on the clinicaltrial.gov was the reduction in migraine days from baseline for the entire study cohort. The result was negative, showing no difference between the two cohorts. The positive result presented in the abstract, is based on the subgroup analysis targeting the episodic migraine subgroup (n=20, 11 for cabergoline vs 9 for placebo), and the result is actually borderline (p=0.04). If the main analysis remains insignificant, the subgroup analysis must be well-founded (better defined a priori) and correct for multiple comparisons, otherwise it is more like data torturing. More importantly, the selective presentation of the subgroup analysis in the abstract is misleading.

**Reviewer #3: ** This paper clarified that cabergoline significantly reduced monthly migraine days in episodic migraine without serious adverse effects, supporting further investigation into the use of cabergoline for migraine prevention.

This reviewer finds the paper is well written, and appropriate for publication after minor correction shown as follows.

Line 87: [13th of June] -> [13th of June 2023]

Line 168: “Forty-five” will be replaced with “Fifty-four”

**Reviewer #4:**  Thank you for inviting me to review this manuscript. I believe the authors have conducted a clinical study that is beneficial for migraine patients. Its quality is sufficient to be published after revisions. Below are my suggestions:

1. Further Discussion on Study Limitations

Although the discussion section mentions the small sample size as a limitation, other potential influencing factors (e.g., short follow-up duration, sample selection bias) should be explored further. Suggestions for addressing these limitations, such as conducting multicenter or larger-scale studies in the future, can also be included.

2. Clarify Titles and Annotations in Figures

The data presented in the figures are central to the manuscript, particularly Figures 2 and 3. Strengthening the descriptions in the figure titles and annotations can facilitate readers' quick understanding while highlighting the key conclusions of the study.

3. Enhance the Rigor of Background Argumentation

The background section mentions that "the relationship between dopamine and migraines remains controversial," but the cited references are not sufficiently comprehensive. Recent findings on the role of dopamine in migraine mechanisms should be incorporated to improve the relevance and credibility of the references.

4. Improve Readability of Statistical Method Descriptions

The statistical analysis mentions the use of linear regression models and non-parametric tests, but the rationale for selecting these methods is not well explained. Additional clarification on why these methods were chosen and how potential confounding factors were controlled would enhance the methodological clarity.

5. Supplement Detailed Analysis of Adverse Events

While the types and frequencies of adverse events are listed, their impact on patients has not been detailed. Adding analyses based on scales or patient feedback to quantify the impact of adverse events on quality of life could strengthen this aspect of the manuscript.

6. Enhance the Title and Abstract to Increase Appeal

The title and abstract are crucial for capturing readers' attention. The current title is straightforward but lacks appeal. It is recommended to highlight terms such as “first evidence” or “significant efficacy” in the title. The abstract should also emphasize the study's innovative contribution compared to previous research.

7. Refine Discussion of Limited Effects on Chronic Migraine

The discussion mentions that cabergoline has limited efficacy for chronic migraine, but the potential mechanism differences (e.g., varying responses to the CGRP pathway) or directions for treatment optimization are not deeply explored. Adding hypotheses or further discussion in this section would add depth. The authors could also enrich this section by citing related articles at appropriate positions in the discussion

(e.g., https://journals.plos.org/plosone/article?id=10.1371/journal.pone.0304370).

8. Improve Data Availability and Ethics Statements

The data sharing statement mentions that sensitive data cannot be fully disclosed but does not elaborate on how other researchers can access the data. It is recommended to provide a more specific data request process in the “Data Availability” section.

**Reviewer #5:**  I thank the authors for testing, in an RCT, the effects of a molecule with significant metabolic effects, acting on the hypothalamic-pituitary axis.

In my opinion, the RCT yielded interesting results.

I suggest that the authors evaluate the effect of cabergoline on glucose metabolism (1, 2) as a potential mechanism underlying the reduction in MMD in episodic migraine (3, 4).

If the authors had monitored weight, BMI, and HOMA-IR at baseline and after the intervention, it would be useful to include these data in the manuscript.

The following point, in my opinion, requires correction:

You wrote: "Cabergoline significantly reduced MMD in episodic migraine and led to a significant global patient perceived improvement in both episodic and chronic migraine," but in the chapter Patient Global Impression of Change you stated: "PGIC significantly improved after cabergoline compared to placebo in the entire group, and in participants with episodic migraine. No significant difference was found in chronic migraine."

So, was there a significant improvement in PGIC in chronic migraine or not?

If my observation is correct, you should also revise the graphical abstract to reflect this aspect.

References

1. Inancli SS, Usluogullari A, Ustu Y, Caner S, Tam AA, Ersoy R, et al. Effect of cabergoline on insulin sensitivity, inflammation, and carotid intima media thickness in patients with prolactinoma. Endocrine. 2013;44:193–9.

2. dos Santos Silva CM, Barbosa FR, Lima GA, Warszawski L, Fontes R, Domingues RC, et al. BMI and metabolic profile in patients with prolactinoma before and after treatment with dopamine agonists. Obes (Silver Spring). 2011;19:800–5.

3. Gross EC, Lisicki M, Fischer D, Sándor PS, Schoenen J. The metabolic face of migraine - from pathophysiology to treatment. Nat Rev Neurol. 2019 Nov;15(11):627-643.

4. Del Moro L, Rota E, Pirovano E, Rainero I. Migraine, Brain Glucose Metabolism and the "Neuroenergetic" Hypothesis: A Scoping Review. J Pain. 2022 Aug;23(8):1294-1317.

**Reviewer #6: ** Thank you for the opportunity to review this manuscript. Overall this paper is clinically relevant and very well written. I have some minor points only.

Introduction: Please remove the last sentence on primary outcome (lines 81-82)

Methods:

- Please write the year when the recruitement ended.

- You excluded patients on dopamine antagonists, however some widely used antiemetics in migraine are dopamine antagonists, eg. metoclopramide and domperidone. Were these medications allowed? If yes, would you perform stratified analyses on patients who took antidopaminergic antiemetics vs those who didn't?

- I couldn’t find any comment on power analysis. On which basis did you decide to include 36 participants only? Please explain. This is important given that, although non-significant, the carbegoline group had a higher reduction of MMD. Might it be that in a bigger population the difference would become significant? I guess that the biologic analgetic effect of placebo in pain studies (which you rightly touch on in the discussion section) makes reaching the statistical significance more difficult, which requires a careful planning.

Discussion:

- When comparing absolute values of MMD reduction from your study to other studies, please be careful, as those results are likely based on populations with different baseline characteristics. I get the point and would not delete that part, but I would make the reader aware, that such comparisons should be interpreted with caution.

- In the part where you discuss the risk of valvulopathy, please clarify whether there is a lack of evidence (eg due to scarcity of experience with the carbegoline), or whether there is evidence of a low risk of valvulopathy. As it reads now, I get the impression that there is no data available, which is different to evidence of a low risk. This is obviously important, because we do not want to gather experience on valvulopathy on migraine patients treated with carbegoline, given that the majority are young females.

**Reviewer #7: ** I have to congratulate the authors to perform such interesting IIT in a time when everyone is talking CGRP only.

The study is in gerenal sound and the manuscript is well written. Knowing that this is a pilot study, it would still be very interesting, if the authors performed a sample size calculation.

Still, as the study has failed the primary outcome, this is a negative study. This has to been stated clearly throughout the whole manuscript and especially in the abstract. After that some secondary endpoints and subgroup findings can be discussed and presented. But this has to be somewhat more conservative. All in all the numbers - especially for the subgroups - are rahter small to draw any conclusion. What is in the end confusing is the fact, that there was a difference in eM of 3.6 MMD and 2.3 in cM, but for the whole group only 1.3 (please explain, this is small number statistics!!)

Minor:

- line 49: what is the reference here. The life-time prevalance of migraine is much higher. 15% corresponds rather to point-prevalence.

- line 168: This should maybe read "Fifty-four", otherwise the sum is not correct

- line 271: this is not fully true, in some studies also RWE effects in cM were much higher

- line 276: if you divert eM and cM in the others you also should do this for your data: 3.6 and 2.3 days (than there is less difference in cM)

Please explain/discuss:

- Line 233: why was there no change in MIDAS/HIT-6

- Line 236: why did so many guess correctly?

- It seems that Cabergoline can not be blinded? Differences in AEs, so many guess correctly, difference in PGIC?!

- Please discuss possible issues of MOH in the long-term when acting and 5HT-receptors!

**Reviewer #8:**  Dear Author,

I think this is a very interesting study that suggests another possibility for migraine prophylaxis, so it is worth publishing. However, I would like to point out several points where more information should be added.

Firstly, when did recruitment end? The 13th of June 2023? If so, please add "2023" to the text (line 87).

The second is about the recorded "migraine" attacks. A migraine day seems to be defined by an electronic diary recorded by the participants themselves. How did the authors confirm that these pain attacks were really migraine attacks? Especially in CM patients, attacks can have several different patterns. Were there no participants with multiple types of headaches, such as tension-type headache?

Also, in line 168-169, "Forty-five patients" should be "Fifty-four patients".

The last point concerns the discussion. Although I agree with the significant efficacy of cabergoline shown in this study, it was still unclear whether this can be generalized beyond the sex difference, as most participants were female. In addition, the authors refer to an interesting change that occurred in female mice, reported in one of the references. Therefore, I think that the suggested efficacy of the present study should be limited to female patients. I will not deny possibility of efficacy in male patients; however, it is not able to refer that in this study. This is one of the limitations of this study.

**Reviewer #9:**  The sample size is too small, the results are not convincing, and the reduction in MMD days alone is not convincing. More pain characteristic needs to be provided, such as pian duration, changes in VAS scores, and changes in accompanying symptoms.

6. PLOS authors have the option to publish the peer review history of their article (what does this mean? ). If published, this will include your full peer review and any attached files.

**Do you want your identity to be public for this peer review?** For information about this choice, including consent withdrawal, please see our Privacy Policy .

Reviewer #1: No

Reviewer #2: No

Reviewer #3: No

Reviewer #4: No

Reviewer #5: **Yes: ** Lorenzo Del Moro

Reviewer #6: No

Reviewer #7: No

Reviewer #8: No

Reviewer #9: No

---

## [Author Response · Author response to Decision Letter 1]

29 Jan 2025

Reviewer #1: The sample size is extremely small and so the study is not qualified as a randomized trial. The results are hardly generalizable. Also results are dominated for female subjects.

Yet from Table 3 we can see a large number of adverse events in the Cabergoline group compared to control group, even for this small study. The drug may not be safely applied.

The results are full of uncertainty. For example, the box plots in Figure 2 are very wide. The chronic migraine subjects even show smaller changes for placebo group than cabergoline group.

We thank the reviewer for taking time to review our manuscript. We agree that the sample size is small as also stated in the discussion as a limitation. However, we disagree that it is not qualified as an RCT since it is randomized, double blind, placebo-controlled, approved by the Ethics Committee system and the National Medicines Agency and conducted in compliance with GCP. Indeed, the majority of the participants were females as are most migraine patients (3:1). We agree that the predominance of female participants limits the generalizability of the results with respect to male patients. This is a limitation that we have now addressed more explicitly in the discussion section (line 357-359):

Finally, the predominance of female participants limits the generalizability of the results, and our findings should therefore be interpreted with caution when applied to male patients.

Adverse effects are known for cabergoline, but we argue that the compound is overall very safe, and this particular (very low) dose has been used for the treatment of hyperprolactinemia for > 30 years. We agree that this is a pilot study and that the placebo effect is sizable, which applies to all migraine studies as also mentioned in our paper. The combination of potentially promising results from this pilot study, the need for additional and affordable migraine treatments, and the safety of cabergoline justifies a new and larger RCT.

Reviewer #2: This is a double-blind, placebo-controlled study that investigated the effect of cabergoline 0.5mg for migraine prevention. There are several key issues with this trial that need to be explained.

1. According to the clinicaltrial.gov online registry, this is supposed to be a study of chronic migraine, which is defined as having more of than 15 headache days per month, of which at least 8 are migrainous headaches. Therefore, the inclusion criteria clearly contradict the topic of this registration, and the current study includes both episodic and chronic migraine.

We appreciate your careful review and the opportunity to clarify this important point regarding our study's registration and inclusion criteria.

The clinicaltrials.gov registry entry for this study originally targeted chronic migraine patients. However, following protocol refinement prior to study initiation, we expanded the inclusion criteria to encompass both episodic and chronic migraine patients. This is reflected in the inclusion criteria of the final version of the protocol approved by the authorities and audited by the GCP unit:

• Migraine and more than 6 days with headache every months

The decision to include both episodic and chronic migraine patients was guided by several considerations. First, cabergoline’s potential efficacy spans across the migraine spectrum, and including a broader patient population allows for a more comprehensive evaluation of its effects. Second, as an investigator-initiated trial, this study was exploratory in nature and aimed to provide preliminary data on the safety and efficacy of cabergoline, thereby laying the groundwork for future larger-scale and more targeted investigations.

To address the heterogeneity in the study population, we conducted subgroup analyses to separately evaluate the effects of cabergoline on episodic and chronic migraine patients.

2. There is insufficient evidence to justify the use of cabergoline as a migraine preventive drug, given the potential side effects of gambling/sex/other urges, and others. The authors did not provide sufficient argument for its use in migraine other than case series of comorbid migraine and hyperprolactinemia. The potential binding affinity of cabergoline to 5-HT1B/1D receptors is not a valid argument, as studies with triptans (5-HT1B/1D agonists) do not provide evidence of prevention but may increase the risk of medication overuse headache.

Thank you for raising these points. It is correct that cabergoline may carry side effects as mentioned. Ludomania and hypersexuality AE are, however, exceedingly rare and the cabergoline dose used is very low and similar to that, which has been used to treat hyperprolactinemia for > 30 years. As regards the rationale, we mention a previous (and promising) study using bromocriptine, which also is a dopamine agonist used to treat hyperprolactinemia. Moreover, elevated prolactin levels have been implicated in migraine pathogenesis.

3. The study cohort is small (18 cabergoline vs. 18 placebo). There is no information on the power calculation and whether the study is sufficiently powered to detect a difference between the two cohorts.

This study was designed as a pilot trial. The sample size (18 participants per arm) reflects the exploratory nature of this investigator-initiated trial. While this cohort size limits the ability to detect subtle differences between the groups, it provides valuable preliminary data to inform the design of larger, adequately powered trials. Indeed, pilot studies in this field are recommended in guidelines from the International Headache Society in order to ‘…provide insights that improve the design of fully-powered studies, including a basis for sample size calculations’ (DOI: 10.1177/0333102420941839).

The small sample size is a limitation, which we also state in the discussion. We have tried to emphasize this further in the revised discussion (line 351-359):

The major strength of our study lies in its randomized, placebo-controlled design, which minimizes bias and enhances validity. However, the small sample size remains a limitation, increasing the risk of random variability and the potential of overestimation or underestimation of effects. ... A larger study is needed to substantiate our findings.

A formal power calculation was not conducted prior to the trial due to the limited existing data on the efficacy of cabergoline for migraine prevention. However, we believe that this pilot study demonstrates the potential of cabergoline in this patient population and provides an important foundation for designing future adequately powered trials.

4. The primary outcome registered on the clinicaltrial.gov was the reduction in migraine days from baseline for the entire study cohort. The result was negative, showing no difference between the two cohorts. The positive result presented in the abstract, is based on the subgroup analysis targeting the episodic migraine subgroup (n=20, 11 for cabergoline vs 9 for placebo), and the result is actually borderline (p=0.04). If the main analysis remains insignificant, the subgroup analysis must be well-founded (better defined a priori) and correct for multiple comparisons, otherwise it is more like data torturing. More importantly, the selective presentation of the subgroup analysis in the abstract is misleading.

Thank you for your feedback regarding the presentation of our results and the subgroup analysis. We appreciate the opportunity to address these concerns.

We fully agree with the importance of avoiding data misrepresentation or "data torturing”, but we disagree that corrections for multiple testing are necessary in this context. The study included only two subgroups (chronic and episodic migraine), and the subgroup analysis is clinically and scientifically sound. As a pilot study, the primary goal was to generate hypotheses and provide preliminary insights as also recommended by the International Headache Society Guidelines (DOI: 10.1177/0333102420941839), and our analysis was consistent with this approach. While the subgroup analysis of chronic versus episodic migraine was not predefined in the protocol, it was guided by the clinical relevance of these subgroups. Chronic and episodic migraines may differ in terms of pathophysiology, disease burden, and potential treatment responses, making this analysis important for understanding the differential effects of cabergoline. We recognize that this limitation should have been stated more clearly in the manuscript, and we have revised the discussion to reflect this point (line 355-357):

… the subgroup analysis in patients with chronic and episodic migraine was not predefined, however it was justified by its clinical relevance.

We agree that the study did not meet its primary endpoint, a reduction in migraine days from baseline for the entire study cohort. This is clearly stated at the top of the discussion; however, we acknowledge that this was missing in the abstract. We have revised the abstract to clearly include the results of the primary outcome, noting that no significant difference was observed in the overall cohort (31-35):

No significant overall difference in the reduction of monthly migraine days was observed. However, among participants with episodic migraine (n= 20), the mean (SE) reduction in monthly migraine days from baseline to the last 28 days of the treatment period was -5.4 (1.3) with cabergoline compared to -1.8 (0.9) with placebo (p=0.04) [odds ratio: 0.79 (95% CI 0.65 – 0.95), p=0.014].

Reviewer #3: This paper clarified that cabergoline significantly reduced monthly migraine days in episodic migraine without serious adverse effects, supporting further investigation into the use of cabergoline for migraine prevention.

This reviewer finds the paper is well written, and appropriate for publication after minor correction shown as follows.

We thank the reviewer for the positive evaluation of our paper including the support to further investigation into the use of cabergoline migraine prevention.

Line 87: [13th of June] -> [13th of June 2023]

Line 168: “Forty-five” will be replaced with “Fifty-four”

Thank you for the thorough proofreading and for identifying these typos. We have revised accordingly.

Reviewer #4: Thank you for inviting me to review this manuscript. I believe the authors have conducted a clinical study that is beneficial for migraine patients. Its quality is sufficient to be published after revisions. Below are my suggestions:

We thank the reviewer for the positive evaluation and not least – for pointing out that it may eventually benefit migraine patients.

1. Further Discussion on Study Limitations

Although the discussion section mentions the small sample size as a limitation, other potential influencing factors (e.g., short follow-up duration, sample selection bias) should be explored further. Suggestions for addressing these limitations, such as conducting multicenter or larger-scale studies in the future, can also be included.

We acknowledge that a longer follow-up period might have provided additional insights into the long-term efficacy, safety, and durability of cabergoline's effects. While a study period of 12 weeks is commonly applied in migraine research, in the revised manuscript, we have emphasized the potential value of extended follow-up periods in future studies (line 359-361):

A larger study is needed to substantiate our findings. In this context, a longer follow-up period may be considered to evaluate long-term efficacy, safety, and durability of cabergoline's effects.

2. Clarify Titles and Annotations in Figures

The data presented in the figures are central to the manuscript, particularly Figures 2 and 3. Strengthening the descriptions in the figure titles and annotations can facilitate readers' quick understanding while highlighting the key conclusions of the study.

Thank you for pointing out the need to clarify the titles and annotations in the Figures. We have edited the legends as suggested in order to improve the readability.

3. Enhance the Rigor of Background Argumentation

The background section mentions that "the relationship between dopamine and migraines remains controversial," but the cited references are not sufficiently comprehensive. Recent findings on the role of dopamine in migraine mechanisms should be incorporated to improve the relevance and credibility of the references.

We appreciate the reviewer’s suggestion to enhance the rigor of the background section. We have revised the text to incorporate recent findings on the role of dopamine in migraine mechanisms (line 62-72):

Growing evidence underscores the importance of dopamine in the pathophysiology of migraine, particularly the dopamine D2 receptor, though its exact role remains ambiguous. Dopamine antagonistic drugs are commonly used for acute migraine treatment and certain premonitory symptoms are considered dopamine-driven [1]. On the other hand, the dopamine D2 receptor agonist bromocriptine, an ergot alkaloid derivative, and other dopamine receptor agonists have demonstrated beneficial effects in migraine management [2, 3]. In vivo PET imaging studies have shown an imbalance in dopamine D2/D3 receptor activity during migraine attacks, indicating fluctuations in endogenous dopamine release that correlate with the chronicity and frequency of migraine attacks [4]. Additionally, studies in rodents suggest that D2 receptor activation may influence central nociceptive sensitization in chronic migraine [5, 6].

4. Improve Readability of Statistical Method Descriptions

The statistical analysis mentions the use of linear regression models and non-parametric tests, but the rationale for selecting these methods is not well explained. Additional clarification on why these methods were chosen and how potential confounding factors were controlled would enhance the methodological clarity.

We appreciate the feedback. In response to the reviewer’s concern, we have simplified the statistical analysis to enhance clarity. We now report means and standard deviations (SDs) for normally distributed variables, and medians with interquartile ranges (IQRs) for non-normally distributed variables. Comparisons between the groups are made using unpaired t-tests for normally distributed variables, and Mann-Whitney U tests for non-normally distributed variables. We have revised the manuscript to reflect this simplified approach and ensure better readability and transparency (line 158-163).

Primary and secondary outcomes were analysed using unpaired t-test and presented as mean ± standard errors (SE), if data were normally distributed. For non-normally distributed data, the Mann Whitney U test was used, and results are reported as medians with interquartile ranges (IQR). Baseline values are presented descriptively as means and standard deviations (SD) for normally distributed variables, or medians with IQR for non-normally distributed variables.

5. Supplement Detailed Analysis of Adverse Events

While the types and frequencies of adverse events are listed, their impact on patients has not been detailed. Adding analyses based on scales or patient feedback to quantify the impact of adverse events on quality of life could strengthen this aspect of the manuscript.

In our study, adverse events were recorded by type and frequency, however, we recognize the value of patient feedback to quantify the impact of adverse events on quality of life. While specific quality-of-life measures related to adverse events were not collected, the Patients’ Global Impression of Change (PGIC) was used to capture patients’ overall perception of treatment impact. Importantly, PGIC improved quite convincingly compared to placebo, suggesting that the perceived benefits of treatment outweighed any negative effects from adverse events. We believe this measure provides meaningful insight into the balance between efficacy and tolerability in this study. We have emphasized this in the revised manuscript (line 296-299):

Cabergoline was well-tolerated, with side effects limited to mild and transient symptoms, including headache, dizziness, fatigue, and constipation. Notably, PGIC improved in the treatment group compared to placebo, indicating that patients perceived the benefits of treat

---

## [Decision Letter · Decision Letter 1]

27 Feb 2025

Cabergoline as a preventive migraine treatment

A randomized clinical pilot trial

PONE-D-24-49788R1

Dear Dr. Hjelholt,

We’re pleased to inform you that your manuscript has been judged scientifically suitable for publication and will be formally accepted for publication once it meets all outstanding technical requirements.

Kind regards,

Sherief Ghozy, M.D.

Academic Editor

PLOS ONE

**Comments to the Author**

1. If the authors have adequately addressed your comments raised in a previous round of review and you feel that this manuscript is now acceptable for publication, you may indicate that here to bypass the “Comments to the Author” section, enter your conflict of interest statement in the “Confidential to Editor” section, and submit your "Accept" recommendation.

Reviewer #1: All comments have been addressed

2. Is the manuscript technically sound, and do the data support the conclusions?

Reviewer #1: Yes

3. Has the statistical analysis been performed appropriately and rigorously? 

Reviewer #1: Yes

4. Have the authors made all data underlying the findings in their manuscript fully available?

Reviewer #1: Yes

5. Is the manuscript presented in an intelligible fashion and written in standard English?

Reviewer #1: Yes

6. Review Comments to the Author

Reviewer #1: This paper can be published as a pilot study. The previous comments are minor. I don't have additional issues with this revision.

7. PLOS authors have the option to publish the peer review history of their article (what does this mean? ). If published, this will include your full peer review and any attached files.

**Do you want your identity to be public for this peer review?** For information about this choice, including consent withdrawal, please see our Privacy Policy .

Reviewer #1: No

---

## [Editor Report · Acceptance letter]

PONE-D-24-49788R1

PLOS ONE

Dear Dr. Hjelholt,

I'm pleased to inform you that your manuscript has been deemed suitable for publication in PLOS ONE. Congratulations! Your manuscript is now being handed over to our production team.

Kind regards,

on behalf of

Dr. Sherief Ghozy

Academic Editor

PLOS ONE